# Symmetry-Aware Actor-Critic for 3D Molecular Design

**Gregor N. C. Simm,**[1] **Robert Pinsler,**[1] **Gábor Csányi,**[1] **& José Miguel Hernández-Lobato**[1,2]
[1]Department of Engineering, University of Cambridge, Cambridge, UK
[2]Alan Turing Institute, London, UK
`{gncs2,rp586,gc121,jmh233}@cam.ac.uk`

## Abstract

Automating molecular design using deep reinforcement learning (RL) has the potential to greatly accelerate the search for novel materials. Despite recent progress on leveraging graph representations to design molecules, such methods are fundamentally limited by the lack of three-dimensional (3D) information. In light of this, we propose a novel actor-critic architecture for 3D molecular design that can generate molecular structures unattainable with previous approaches. This is achieved by exploiting the symmetries of the design process through a rotationally covariant state-action representation based on a spherical harmonics series expansion. We demonstrate the benefits of our approach on several 3D molecular design tasks, where we find that building in such symmetries significantly improves generalization and the quality of generated molecules.

## 1 Introduction

The search for molecular structures with desirable properties is a challenging task with important applications in *de novo* drug design and materials discovery (Schneider et al., 2019). There exist a plethora of machine learning approaches to accelerate this search, including generative models based on variational autoencoders (VAEs) (Gómez-Bombarelli et al., 2018), recurrent neural networks (RNNs) (Segler et al., 2018), and generative adversarial networks (GANs) (De Cao & Kipf, 2018). However, the reliance on a sufficiently large dataset for exploring unknown regions of chemical space is a severe limitation of such supervised models. Recent RL-based methods (e.g., Olivecrona et al. (2017), Jørgensen et al. (2019), Simm et al. (2020)) mitigate the need for an existing dataset of molecules as they only require access to a reward function.

Most approaches rely on graph representations of molecules, where atoms and bonds are represented by nodes and edges, respectively. This is a strongly simplified model designed for the description of single organic molecules. It is unsuitable for encoding metals and molecular clusters as it lacks information about the relative position of atoms in 3D space. Further, geometric constraints on the design process cannot be included, e.g. those given by the active site of an enzyme. A more general representation closer to the physical system is one in which a molecule is described by its atoms' positions in Cartesian coordinates. However, it would be very inefficient to naively learn a model based on this representation. That is because molecular properties such as the energy are invariant (i.e. unchanged) under symmetry operations like translation or rotation of all atomic positions. A model without the right inductive bias would thus have to learn those symmetries from scratch.

In this work, we develop a novel RL approach for designing molecules in Cartesian coordinates that explicitly encodes these symmetry operations. The agent builds molecules by consecutively placing atoms such that if the generated structure is rotated or translated, the agent's action is rotated and translated accordingly; this way, the reward remains the same (see Fig. 1 (a)). We achieve this through a rotationally *covariant* state representation based on spherical harmonics, which we integrate into a novel actor-critic network architecture with an auto-regressive policy that maintains the desired covariance. Building in this inductive bias enables us to generate molecular structures with more complex coordination geometry than the class of molecules that were attainable with previous approaches. Finally, we perform experiments on several 3D molecular design tasks, where

we find that our approach significantly improves the generalization capabilities of the RL agent and the quality of the generated molecules.

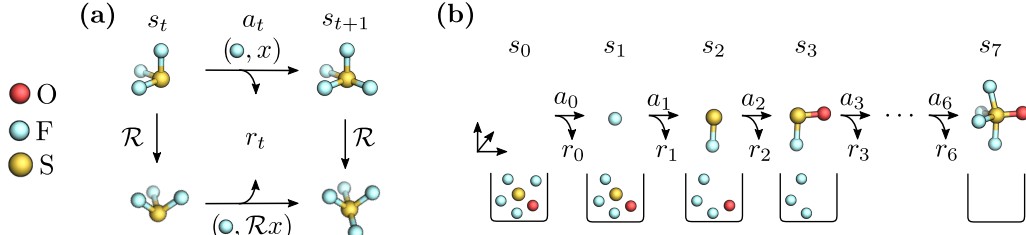

Figure 1: **(a)** Illustration of a rotation-covariant state-action representation. If the structure is rotated by $\mathcal{R}$, the position $x$ of the action transforms accordingly. **(b)** Rollout with bag $\mathcal{B}_0 = \mathrm{SOF}_4$. The agent builds a molecule by repeatedly taking atoms from the bag and placing them onto the 3D canvas. Bonds connecting atoms are only for illustration and not part of the MDP.

In summary, our contributions are as follows:

- we propose the first approach for 3D molecular design that exploits symmetries of the design process by leveraging a rotationally covariant state representation;
- we integrate this state representation into an actor-critic neural network architecture with a rotationally covariant auto-regressive policy, where the orientation of the atoms to be placed is modeled through a flexible distribution based on spherical harmonics;
- we demonstrate the benefits of our approach on several 3D molecular design tasks, including a newly proposed task that showcases the generalization capabilities of our agent.

## 2 BACKGROUND

### 2.1 REINFORCEMENT LEARNING FOR MOLECULAR DESIGN

In the standard RL setting (Sutton & Barto, 2018), an agent interacts with the environment to maximize its reward. Formally, such an environment is described by a Markov decision process (MDP) $\mathcal{M} = (\mathcal{S}, \mathcal{A}, \mathcal{T}, \mu_0, \gamma, r)$ with states $s_t \in \mathcal{S}$, actions $a_t \in \mathcal{A}$, transition dynamics $\mathcal{T} : \mathcal{S} \times \mathcal{A} \mapsto S$, initial state distribution $\mu_0$, discount factor $\gamma \in (0, 1]$, and reward function $r : \mathcal{S} \times \mathcal{A} \mapsto \mathbb{R}$. The goal is to learn a stochastic policy $\pi(a_t|s_t)$ that maximizes the expected discounted return $J(\theta) = \mathbb{E}_{s_0 \sim \mu_0}[V^\pi(s_0)]$, where the value function $V^\pi(s_t) = \mathbb{E}_\pi[\sum_{t'=t}^T \gamma^{t'} r(s_{t'}, a_{t'})|s_t]$ is defined as the expected discounted return when starting from state $s_t$ and following policy $\pi$.

Following Simm et al. (2020), we design molecules by iteratively picking atoms from a **bag** and positioning them on a 3D **canvas**. Such a sequential decision-making problem is described by an MDP where the state $s_t = (\mathcal{C}_t, \mathcal{B}_t)$ comprises both the canvas $\mathcal{C}_t$ and the bag $\mathcal{B}_t$. The canvas $\mathcal{C}_t = \mathcal{C}_0 \cup \{(e_i, x_i)\}_{i=0}^{t-1}$ is a set of atoms with chemical element $e_i \in \{\mathrm{H, C, N, O}, \dots\}$ and position $x_i \in \mathbb{R}^3$ placed up to time $t - 1$, where $\mathcal{C}_0$ can either be empty or contain a set of initially placed atoms. The number of atoms on the canvas is denoted by $|\mathcal{C}_t|$. The bag $\mathcal{B}_t = \{(e, m(e))\}$ is a multi-set of atoms yet to be placed, where $m(e)$ is the multiplicity of the element $e$. Each action $a_t = (e_t, x_t)$ consists of the element $e_t \in \mathcal{B}_t$ and position $x_t \in \mathbb{R}^3$ of the next atom to be added to the canvas. Placing an atom through action $a_t$ in state $s_t$ is modeled by a deterministic transition function $\mathcal{T}(s_t, a_t)$ that yields the next state $s_{t+1} = (\mathcal{C}_{t+1}, \mathcal{B}_{t+1})$ with $\mathcal{B}_{t+1} = \mathcal{B}_t \backslash e_t$.

The reward function $r(s_t, a_t) = -\Delta E(s_t, a_t)$ is given by the negative energy difference between the resulting structure described by $\mathcal{C}_{t+1}$, and the sum of energies of the current structure $\mathcal{C}_t$ and a new atom of element $e_t$ placed at the origin, i.e. $\Delta E(s_t, a_t) = E(\mathcal{C}_{t+1}) - [E(\mathcal{C}_t) + E(\{(e, \mathbf{0})\})]$. Intuitively, the reward encourages the agent to build stable, low-energy structures. We evaluate the energy using the fast semi-empirical Parametrized Method 6 (PM6) (Stewart, 2007) as implemented in SPARROW (Husch et al., 2018; Bosia et al., 2020); see Appendix A for details.

An example of a rollout is shown in Fig. 1 (b). At the beginning of the episode, the agent observes the initial state $(\mathcal{C}_0, \mathcal{B}_0) \sim \mu_0(s_0)$, e.g. $\mathcal{C}_0 = \emptyset$ and $\mathcal{B}_0 = \text{SOF}_4$[1]. The agent then iteratively constructs a molecule by placing atoms from the bag onto the canvas until the bag is empty.[2]

## 2.2 ROTATIONALLY COVARIANT NEURAL NETWORKS

A function $f : \mathcal{X} \mapsto \mathcal{Y}$ is **invariant** under a transformation operator $T_g : \mathcal{X} \mapsto \mathcal{X}$ if $f(T_g[x]) = f(x)$ for all $x \in \mathcal{X}, g \in G$, where $G$ is a mathematical group. In contrast, $f$ is **covariant** with respect to $T_g$ if there exists an operator $T'_g : \mathcal{Y} \mapsto \mathcal{Y}$ such that $f(T_g[x]) = T'_g[f(x)]$. To achieve rotational covariance, it is natural to work with spherical harmonics. They are a set of complex-valued functions $Y_\ell^m : \mathbb{S}^2 \mapsto \mathbb{C}$ with $\ell = 0, 1, 2, \dots$ and $m = -\ell, -\ell + 1, \dots, \ell - 1, \ell$ on the unit sphere $\mathbb{S}^2$ in $\mathbb{R}^3$. The first few spherical harmonics are in Appendix B. They are defined by

$$Y_\ell^m(\vartheta, \varphi) = (-1)^m \sqrt{\frac{2\ell + 1}{4\pi} \frac{(\ell - m)!}{(\ell + m)!}} P_l^m(\cos(\vartheta)) e^{im\varphi}, \quad \varphi \in [0, 2\pi], \quad \vartheta \in [0, \pi], \quad (1)$$

where $P_\ell^m$ denotes the associated normalized Legendre polynomials of the first kind (Bateman, 1953), and each $Y_\ell^m$ is normalized such that $\int\int |Y_\ell^m(\vartheta, \varphi)|^2 \sin \vartheta d\vartheta d\varphi = 1$. Any square-integrable function $f : \mathbb{S}^2 \mapsto \mathbb{C}$ can be written as a series expansion in terms of the spherical harmonics,

$$f(\tilde{x}) = \sum_{\ell=0}^{\infty} \sum_{m=-\ell}^{\ell} \hat{f}_\ell^m Y_\ell^m(\tilde{x}), \quad (2)$$

where $\tilde{x} = (\vartheta, \varphi) \in \mathbb{S}^2$. The complex-valued coefficients $\{\hat{f}_\ell^m\}$ are the analogs of Fourier coefficients and are given by $\hat{f}_\ell^m = \int f(\tilde{x}) Y_\ell^{m*}(\tilde{x}) \Omega(d\tilde{x})$. Such a function $f$ can be modeled by learning the coefficients $\{\hat{f}_\ell^m\}$ using CORMORANT (Anderson et al., 2019), a neural network architecture for predicting properties of chemical systems that works entirely in Fourier space. A key feature is that each neuron is covariant to rotation but invariant to translation; further, each neuron explicitly corresponds to a subset of atoms in the molecule. The input of CORMORANT is a spherical function $f^0 : \mathbb{S}^2 \mapsto \mathbb{C}^d$ and the output is a collection of vectors $\hat{f} = \{\hat{f}_0, \hat{f}_1, \dots, \hat{f}_L\}$, where each $\hat{f}_\ell \in \tau \times (2\ell + 1)$ is a rotationally covariant vector with $\tau$ channels. That is, if the input is rotated by $\mathcal{R} \in \text{SO}(3)$, then each $\hat{f}_\ell$ transforms as $\hat{f}_\ell \mapsto D^\ell(\mathcal{R})\hat{f}_\ell$, where $D^\ell(\mathcal{R}) : \text{SO}(3) \mapsto \mathbb{C}^{(2\ell+1)\times(2\ell+1)}$ are the irreducible representations of $\text{SO}(3)$, also called the Wigner D-matrices.

## 3 COVARIANT POLICY FOR MOLECULAR DESIGN

An efficient RL agent needs to exploit the symmetries of the molecular design process. Therefore, we require a policy $\pi(a|s)$ with actions $a = (e, x)$ that is covariant under translation and rotation with respect to the position $x$, i.e., $x$ should rotate (or translate) accordingly if the atoms on the canvas $\mathcal{C}$ are rotated (or translated). In contrast, the policy needs to be invariant to the element $e$, i.e. the chosen element remains unchanged under such transformations (see Fig. 1 (a)). Since learning such a policy is difficult when working directly in global Cartesian coordinates, we instead follow Simm et al. (2020) and use an action representation that is local with respect to an already placed **focal atom**. If the next atom is placed relative to the focal atom, covariance under translation of $x$ is automatically achieved and only the rotational covariance remains to be dealt with.

As shown in Fig. 2, we model the action $a$ through a sequence of sub-actions: (1) the index $f \in \{1, \dots, |\mathcal{C}|\}$ of the focal atom around which the next atom is placed,[3] (2) the element $e \in \{1, \dots, N_e\}$ of the next atom from the set of available elements, (3) a distance $d \in \mathbb{R}_+$ between the focal atom and the next atom, and (4) the orientation $\tilde{x} = (\vartheta, \varphi) \in \mathbb{S}^2$ of the atom on a unit sphere around the focal atom. Denoting $x_f$ as the position of the focal atom, we obtain action $a = (e, x)$ by mapping the local coordinates $(\tilde{x}, d, f)$ to global coordinates $x = x_f + d \cdot \tilde{x}$, where

---

[1]Shorthand for $\{(S, 1), (O, 1), (F, 4)\}$.

[2]Hereafter, we drop the time index when it is clear from the context.

[3]If the canvas $\mathcal{C}_0$ is empty, the agent selects an element $e_0 \in \mathcal{B}_0$ and places it at the origin, i.e. $a_0 = (e_0, \mathbf{0})$.

$x$ is now covariant under translation *and* rotation. We choose these sub-actions using the following auto-regressive policy:

$$\pi(a|s) = \pi(\tilde{x}, d, e, f|s) = p(\tilde{x}|d, e, f, s)\, p(d|e, f, s)\, p(e|f, s)\, p(f|s). \tag{3}$$

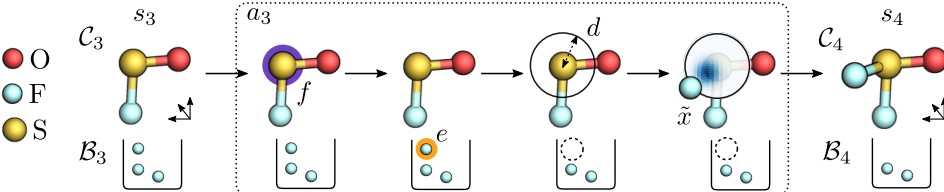

Figure 2: Action representation of the auto-regressive policy. The agent chooses focal atom $f$, element $e$, distance $d$, and orientation $\tilde{x}$. We then map back to global coordinates $x$ to obtain action $a = (e, x)$. Bonds between atoms are only for illustration.

A novel actor-critic neural network architecture that implements this policy is illustrated in Fig. 3. In the following, we discuss its state embedding, actor, and critic networks in more detail.

## 3.1 State Embedding

The state embedding network transforms canvas $\mathcal{C}$ and bag $\mathcal{B}$ to obtain a rotationally covariant and translationally invariant representation. For that, we concatenate a vectorized representation of the bag with each atom on the canvas and feed it into CORMORANT, i.e. $s^{\mathrm{cov}} \leftarrow \mathrm{CORMORANT}(\mathcal{C}, \mathcal{B})$, where $s^{\mathrm{cov}} = \{s_\ell^{\mathrm{cov}}\}_{\ell=0}^{L_{\max}}$, $s_\ell^{\mathrm{cov}} \in \mathbb{C}^{|\mathcal{C}| \times \tau \times (2\ell+1)}$, and $\tau$ is the number of channels. For the sake of exposition, we assume a single channel for each element in the bag, i.e. $\tau = N_e$ (cf. Fig 3); in practice, we use up to four channels per element.

However, not every sub-action in Eq. (3) should transform equally under rotation and translation. While the orientation $\tilde{x}$ needs to be *covariant* under rotation, the choice of focal atom $f$, element $e$, and distance $d$ have to be *invariant* to rotation and translation. For these sub-actions, we additionally require an invariant state representation. To obtain such a representation $s^{\mathrm{inv}} \in \mathbb{R}^{|\mathcal{C}| \times k}$, we employ a combination of transformations from Anderson et al. (2019) as listed in Appendix C (e.g. for $\ell = 0$, one can simply select the $s_{\ell=0}^{\mathrm{cov}}$ component), which we collectively denote as $\mathcal{T}_{\mathrm{inv}}$.

## 3.2 Actor

**Focal Atom and Element** The distribution $p(f|s)$ over the focal atom $f$ is modeled as categorical, $f \sim \mathrm{Cat}(f; h_f)$, where $h_f$ are the logits for each atom in $\mathcal{C}$ predicted by a multi-layer perceptron (MLP). Likewise, the distribution over the element $e$ is given by $p(e|f, s) = \mathrm{Cat}(e; h_e)$ with $h_e = \mathrm{MLP}_e(s_f^{\mathrm{inv}})$, where $s_f^{\mathrm{inv}}$ is the invariant representation for the focal atom. Since the number of possible focal atoms $f$ increases and the set of available elements $e$ decreases during a rollout, we mask out invalid focal atoms $f \notin \{1, \ldots, |\mathcal{C}_t|\}$ and elements $e \notin \mathcal{B}_t$ by setting their probabilities to zero and re-normalizing the categorical distributions. The agent does *not* make use of chemical concepts like bond connectivity to aid the choice of the focal atom.

**Distance** We select the channel $\tau_e$ corresponding to element $e$ from $s_f^{\mathrm{cov}}$ to obtain $s_{f,e}^{\mathrm{cov}} := \{s_{\ell_{f,e}}^{\mathrm{cov}}\}_{\ell=0,\ldots,L_{\max}}$ and $s_{f,e}^{\mathrm{inv}} \leftarrow \mathcal{T}_{\mathrm{inv}}(s_{f,e}^{\mathrm{cov}})$. Then we model the distribution over the distance $d$ between the focal atom and the next atom to be placed as a mixture of $M$ Gaussians, $p(d|e, f, s) = \sum_{m=1}^{M} \pi_m \mathcal{N}(\mu_m, \sigma_m^2)$, where $\pi_m$ is the mixing coefficient of the $m$-th Gaussian $\mathcal{N}(\mu_m, \sigma_m^2)$. The mixing coefficients and the means are predicted by a mixture density network (MDN) (Bishop, 1994), i.e. $\{\pi_m, \mu_m\}_{m=1}^{M} = \mathrm{MDN}(s_{f,e}^{\mathrm{inv}})$. The standard deviations $\{\sigma_m\}_{m=1}^{M}$ are global parameters. We guarantee that the sampled distances are positive by clipping values below zero.

**Combining Invariant and Covariant Features** The choice of distance $d$ can significantly affect the orientation $\tilde{x}$ of the atom. For example, if $d$ has the length of a triple bond, then $\tilde{x}$ will be very different from if it was a single bond. Thus, we condition $s_{f,e}^{\mathrm{cov}}$ on distance $d$ through a non-linear and learnable transformation that preserves rotational covariance. We then use this representation

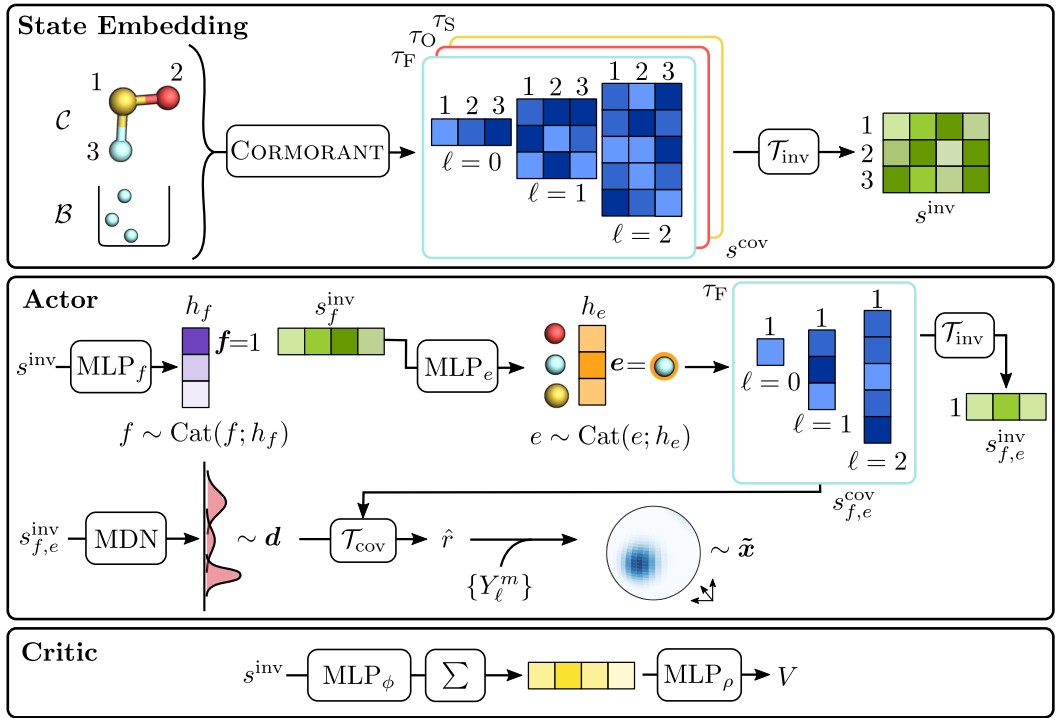

Figure 3: Illustration of the state embedding, actor, and critic networks. Both canvas $\mathcal{C}$ and bag $\mathcal{B}$ are fed to the **state embedding** network CORMORANT to obtain rotation-covariant ($s^{\text{cov}}$) and -invariant ($s^{\text{inv}}$) state representations. The **actor** network then samples the different sub-actions highlighted in bold. The **critic** takes the invariant representation $s^{\text{inv}}$ to compute a value $V$.

to model a spherical distribution over $\tilde{x}$. Kondor & Trivedi (2018) showed that a linear transformation with learnable parameters is only covariant if the operation combines fragments with the same $\ell$. Further, the Clebsch-Gordan (CG) non-linearity allows one to combine two covariant features such that the result is still covariant. Thus, we obtain a rotationally covariant representation $\hat{r} := \{\hat{r}_\ell\}_{\ell=0,\dots,L_{\max}} \leftarrow \mathcal{T}_{\text{cov}}(d, s^{\text{cov}}_{f,e})$ conditioned on all previous sub-actions as follows:

$$\hat{r}_\ell = \left[ s^{\text{cov}}_{\ell_{f,e}} \oplus d \cdot s^{\text{cov}}_{\ell_{f,e}} \oplus (d \cdot s^{\text{cov}}_{f,e} \otimes_{\text{cg}} d \cdot s^{\text{cov}}_{f,e})_\ell \right] \cdot W_\ell \quad \forall \ell, \tag{4}$$

where $\oplus$ denotes the appropriate concatenation of matrices, and $W_\ell$ is a learnable complex-valued matrix. As in Anderson et al. (2019), we perform the CG product $\otimes_{\text{cg}}$ only channel-wise to reduce computational complexity.

**Orientation** Next, we utilize $\hat{r}$ to obtain a rotationally covariant spherical distribution for the orientation $\tilde{x}$ based on the series expansion in Eq. (2). Taking inspiration from commonly used distributions (Jammalamadaka & Terdik, 2019), we propose to use the following expression:

$$p(\tilde{x}|d, e, f, s) = \frac{1}{Z} \exp\left( -\beta \left| \frac{1}{\sqrt{k}} \sum_{\ell=0}^{L_{\max}} \sum_{m=-\ell}^{\ell} \hat{r}_\ell^m Y_\ell^m(\tilde{x}) \right|^2 \right), \tag{5}$$

where $\beta \in \mathbb{R}$ is a scaling parameter, and the term $1/\sqrt{k}$ with $k = \sum_{\ell=0}^{L_{\max}} \sum_{m=-\ell}^{\ell} |\hat{r}_\ell^m|^2$ regularizes the distribution so that it does not approach a delta function. The normalization constant $Z$ is estimated via Lebedev quadrature (Lebedev, 1975; 1977). We sample from the distribution in Eq. (5) using rejection sampling (Bishop, 2009) with a uniform proposal distribution $q(\tilde{x}) = (4\pi)^{-1}$. Note that in contrast to more commonly used parametric distributions (e.g. von Mises-Fisher), this formulation allows to model multi-modalities. We discuss alternatives to Eq. (5) in Appendix D.

### 3.3 CRITIC

The critic needs to compute a value $V$ for the state $s$ that is invariant under translation and rotation. Given $s^{\text{inv}}$, we apply a permutation-invariant set encoding (Zaheer et al., 2017) of the atoms, i.e.

$$V(s) = \text{MLP}_\rho \left( \sum_{i=1}^{|\mathcal{C}|} \text{MLP}_\phi(s_i^{\text{inv}}) \right). \tag{6}$$

Finally, we use PPO (Schulman et al., 2017) to learn the parameters of the actor-critic architecture. To encourage sufficient exploration, we add an entropy regularization term over the categorical sub-action distributions of the auto-regressive policy. For offline evaluation, we evaluate the policy without any exploration by choosing the most probable action. While the mode of the distributions for $f$ and $e$ is available in closed form, we approximate the global mode of the distributions over $d$ and $\tilde{x}$ by evaluating the density at $S$ samples and picking the one with the highest density.

## 4 RELATED WORK

**Reinforcement Learning for Molecular Design** There exists a large variety of RL-based approaches for molecular design using either string- or graph-based representations of molecules (Olivecrona et al., 2017; Guimaraes et al., 2018; Putin et al., 2018; Neil et al., 2018; Popova et al., 2018; You et al., 2018; Zhou et al., 2019). However, the choice of representation limits the molecules that can be generated to a (small) region of chemical space for which the representation is applicable, i.e., single organic molecules. Such representations also prohibit the use of reward functions based on quantum-mechanical properties; instead, heuristics are often used. Lastly, geometric constraints on the design process cannot be imposed as the representation does not include any 3D information.

**Molecular Design in Cartesian Coordinates** Another downside of string- and graph-based approaches is their neglect of information encoded in the interatomic distances. In light of this, Gebauer et al. (2018; 2019) proposed a supervised generative neural network for sequentially placing atoms in Cartesian coordinates. While the model respects local symmetries by construction, atoms are placed on a 3D grid. Similar to other supervised approaches, one further requires a dataset that covers the particular class of molecules to be generated. Hammer and coworkers (Jørgensen et al., 2019; Meldgaard et al., 2020) employed a Deep Q-Network (Mnih et al., 2015) to build planar compounds and crystalline surfaces by placing atoms on a grid. Recently, Simm et al. (2020) presented an RL formulation for molecular design in continuous 3D space. The agent models the position of the next atom to be placed in internal coordinates—i.e. the distance, angle, and dihedral angle with respect to already existing atoms—which are invariant under translation and rotation. By mapping from internal to Cartesian coordinates, they then obtain a policy that is covariant under these symmetry operations. However, as shown in Fig 4, the angle and dihedral angle are only defined with respect to two reference points, which are chosen to be the two closest points to a focal atom. In highly symmetric states, e.g. as commonly encountered in materials, this representation fails to distinguish different configurations as one cannot uniquely select the two closest atoms as reference points anymore. In contrast, we do not rely on such reference points as the agent directly samples the orientation from a spherical distribution.

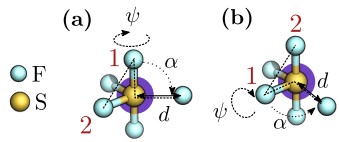

Figure 4: Example of two configurations (a) and (b) that the agent by Simm et al. (2020) cannot distinguish. While the values for distance $d$, angle $\alpha$ and dihedral angle $\psi$ are the same, choosing different reference points (in red) leads to a different action. This is particularly problematic in symmetric states, where one cannot uniquely determine these reference points.

**Covariant Neural Networks in Chemical Science** Prior work employed rotationally covariant neural networks to predict translation- and rotation-invariant physical properties (Thomas et al., 2018; Kondor et al., 2018; Weiler et al., 2018; Anderson et al., 2019; Miller et al., 2020; Finzi et al., 2020; Fuchs et al., 2020), e.g. scalars such as the electronic energy. In contrast, we propose a translation-invariant and rotation-covariant neural network architecture for generating molecules. For a more general treatment of covariance (or equivariance) in RL, see van der Pol et al. (2020).

## 5 EXPERIMENTS

We perform experiments to answer the following questions: (1) is the agent able to learn how to build highly symmetric molecules in Cartesian coordinates from scratch, (2) can we increase the validity, diversity, and stability of generated molecules, and (3) does our approach lead to improved generalization? We address (1) and (2) by evaluating the agent on a diverse range of tasks from the MOLGYM benchmark suite (Simm et al., 2020), and (3) on a newly proposed *stochastic-bag* task (see Section 5.1) where bags are sampled from a distribution over bags. In Appendix G, we show with an additional experiment that the agent can learn to place water molecules around a given solute to form a solvation shell.[4]

We compare our approach (COVARIANT) against the RL agent proposed by Simm et al. (2020), which iteratively builds molecules on a 3D canvas by working in internal coordinates (INTERNAL). As an additional baseline, we consider a classical, optimization-based agent (OPT) with access to a black-box function that yields the energy $E(\mathcal{C})$ and the atomic forces $F(\mathcal{C})$ for a given canvas.[5] The agent constructs molecules by alternating between randomly placing an atom and optimizing the structure. Moreover, the agent applies several heuristics inspired by fundamental chemical concepts to guide the placement of atoms. To make the comparisons fair, we grant OPT a comparable computational budget in terms of the total number of energy computations. Finally, for some experiments, the best possible performance based on quantum-chemical calculations can be reported. See Appendices E and F for more details on the baselines and experimental settings, and Appendix H for an additional runtime comparison between the agents.

### 5.1 STOCHASTIC-BAG TASK

In Simm et al. (2020), a set of molecular design tasks was introduced: the *single-bag* task assesses an agent's ability to build single stable molecules, whereas the *multi-bag* task focuses on building several molecules of different composition and size at the same time. A limitation of these tasks is that the initial bags were selected such that they correspond to known formulas, which in practice might not be known *a priori*. In the *stochastic-bag* task, we relax this assumption by sampling from a more general distribution over bags. Before each episode, we construct a bag $\mathcal{B} = \{(e, m(e))\}$ by sampling counts $(m(e_1), ..., m(e_{\max})) \sim \text{Mult}(\zeta, p_e)$, where the bag size $\zeta$ is sampled uniformly from the interval $[\zeta_{\min}, \zeta_{\max}]$. Here, we obtain an empirical distribution $p_e$ from the multiplicities $m(e)$ of a given bag $\mathcal{B}^*$. For example, with $\mathcal{B}^* = \{(\text{H}, 2), (\text{O}, 1)\}$ we obtain $p_{\text{H}} = \frac{2}{3}$ and $p_{\text{O}} = \frac{1}{3}$. Since sampled bags might no longer correspond to valid molecules when placed completely, we discard bags where the sum of valence electrons over all atoms contained in the bag is odd. This ensures that the agent can build a *closed-shell* system.

### 5.2 RESULTS

**Building Highly Symmetric Molecules**    First, we evaluate the ability to build stable molecules featuring high symmetry and coordination numbers (e.g. trigonal bipyramidal, square pyramidal, and octahedral) on the *single-bag* task with bags $\text{SOF}_4$, $\text{IF}_5$, $\text{SOF}_6$, and $\text{SF}_6$. As shown in Fig. 5 (a), COVARIANT can solve the task for $\text{SOF}_4$ and $\text{IF}_5$ within $30\,000$ to $40\,000$ steps, whereas INTERNAL fails to build low-energy configurations as it cannot distinguish highly symmetric intermediates (cf. Fig. 4). Further results in Fig. 5 (b) for $\text{SOF}_6$ and $\text{SF}_6$ show that COVARIANT is capable of building such structures. Likewise, OPT found the optimal structures for all four bags. While the constructed molecules are small in size, they would be unattainable with graph- or string-based methods as such representations lack important 3D information. For example, RDKIT (Landrum, 2019), a state-of-the-art library for 3D structure generation of organic molecules, failed at this task.

**Validity, Diversity, and Stability of Generated Molecules**    Since string and graph representations are not well-suited for designing molecules with complex 3D structure, it is difficult to directly compare to most prior work. To still enable comparisons, we follow the *GuacaMol* benchmark (Brown et al., 2019) and report the chemical validity, diversity, and stability of the molecules generated by the agents for different experiments. A generated structure is considered valid if it can be successfully converted to a molecular graph by the tool XYZ2MOL (Jensen, 2019; Kim & Kim,

---

[4]Source code of the agent and environment is available at `https://github.com/gncs/molgym`.
[5]For the calculation of $E(\mathcal{C})$ and $F(\mathcal{C})$ we employ PM6; the same method as in the reward function.

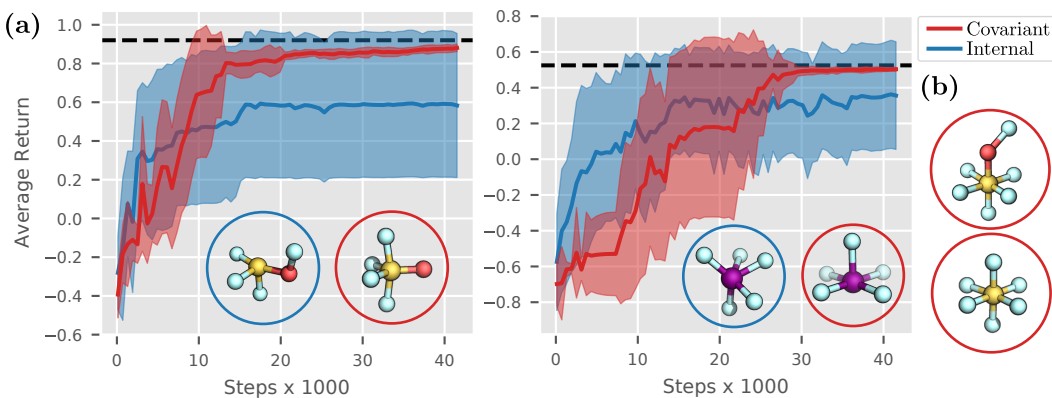

Figure 5: **(a)** Average offline performance on the *single-bag* task with bags $SOF_4$ (left) and $IF_5$ (right) across 10 seeds. In the lower right, molecular structures generated by the agents are shown. Dashed lines denote the optimal return for each experiment. Error bars show two standard deviations. **(b)** Further molecular structures generated by COVARIANT, namely $SOF_6$ and $SF_6$.

Table 1: Validity, diversity, and stability (RMSD in Å) of generated structures.

| Task | Experiment | Validity (↑ is better) | | | Diversity (↑) | | | RMSD (↓) | |
|---|---|---|---|---|---|---|---|---|---|
| | | OPT | INTERNAL | COVARIANT | OPT | INTERNAL | COVARIANT | INTERNAL | COVARIANT |
| Single-bag | $C_3H_5NO_3$ | 0.06 | 0.70 | 0.90 | 19 | 35 | 65 | 0.32 | 0.30 |
| | $C_4H_7N$ | 0.10 | 0.80 | 0.70 | 35 | 18 | 25 | 0.26 | 0.29 |
| | $C_3H_8O$ | 0.06 | 0.90 | 0.80 | 2 | 4 | 8 | 0.42 | 0.22 |
| | $C_7H_{10}O_2$ | 0.05 | 0.50 | 0.80 | 10 | 21 | 85 | 0.80 | 0.76 |
| | $C_7H_8N_2O_2$ | 0.03 | 0.60 | 0.70 | 5 | 58 | 118 | 0.57 | 0.52 |
| Multi-bag | | 0.54 | 0.78 | 0.89 | 22 | 19 | 42 | 0.04 | 0.04 |
| Stochastic-bag | $C_7H_{10}O_2$ | 0.05 | 0.40 | 0.60 | 10 | 26 | 59 | 0.65 | 0.71 |
| | $C_7H_8N_2O_2$ | 0.03 | 0.10 | 0.80 | 5 | 28 | 84 | 0.95 | 0.88 |
| Stochastic-bag (gen.) | $C_7H_{10}O_2$ | 0.00 | 0.00 | 0.13 | 0 | 38 | 68 | n/a | 1.02 |
| | $C_7H_8N_2O_2$ | 0.00 | 0.05 | 0.30 | 0 | 40 | 227 | 1.24 | 1.15 |

2015). The *validity* reported in Table 1 is the ratio of valid molecules generated during offline evaluation at the end of training over 10 seeds. Two molecules are considered identical if their molecular graphs yield the same SMILES strings under RDKIT. The *diversity* shown in Table 1 is the total number of unique and valid structures generated during offline evaluation during training over 10 seeds.[6] In the two *stochastic-bag* experiments, the agents are trained on bags of sizes from the interval $[16, 22]$ sampled with $\mathcal{B}^* = C_7H_8N_2O_2$ and $C_7H_{10}O_2$, respectively. Finally, to assess the *stability* of the generated molecules, valid structures generated in the last iteration underwent a structure optimization using the PM6 method (see Appendix A for details). Then, the root-mean-square deviation of atomic positions (RMSD, in Å) between the original and the optimized structure was computed. In Table 1, the median RMSD is given per experiment.

Results are listed in Table 1. We observe that COVARIANT significantly outperforms the other agents on most experiments both in terms of validity and diversity. The difference is particularly large for the more challenging *stochastic-bag* tasks, where COVARIANT does similarly well as on the *single-bag* experiments. This finding is confirmed in Fig. 6 (a), showing that the exact stoichiometry does not need to be known *a priori* for the agent to build valid molecules. Moreover, the structures generated by COVARIANT are overall slightly more stable compared to INTERNAL. In contrast, OPT often fails to build valid structures. Inspection of the generated structures reveals that for larger bags the agent tends to build multi-molecular clusters, which are considered invalid in this experiment. The stability for OPT is omitted as all of its valid structures are stable by definition.

Compared to graph-based approaches (e.g., Jin et al. (2017); Bradshaw et al. (2019a); Li et al. (2018b;a); Liu et al. (2018); De Cao & Kipf (2018); Bradshaw et al. (2019b)), the average validity

---

[6]For a fairer comparison in the *stochastic-bag* task, we use the structures generated during offline evaluation, instead of those generated during training as in Simm et al. (2020).

and diversity achieved by COVARIANT are still relatively low. This can partly be explained by the fact that state-of-the-art graph-based approaches have the strict rules of chemical bonding in organic molecules encoded into their models. But as a result, they are limited to generating single organic molecules and cannot build molecules for which these rules do not apply (e.g., hypervalent iodine compounds such as $IF_5$). In terms of stability, the supervised generative model by Gebauer et al. (2019) reported an average RMSD of approximately $0.25$ Å. While their approach and the considered molecules are significantly different from ours, this suggests that the generated structures are more stable compared to COVARIANT. Nonetheless, the RL approach presented in this work remains particularly attractive if no dataset exists on which such a supervised model can be trained.

**Generalization** To evaluate the generalization capabilities of all agents to unseen bags, we train on a distribution over bags with $\mathcal{B}^* = C_7H_{10}O_2$ and $C_7H_8N_2O_2$, and test on sets of larger, out-of-distribution bags $\{C_6H_{14}O_3, C_7H_{16}O, C_7H_{16}O_2, C_8H_{18}O\}$, and $\{C_8H_{12}N_2O, C_6H_{12}N_2O_3, C_7H_{14}N_2O, C_7H_{14}N_2O_2\}$ respectively. As shown in Fig. 6 (b), COVARIANT obtains higher average returns with lower variance compared to INTERNAL on $C_7H_{10}O_2$, while performing only slightly worse than the agent trained directly on the test bags (purple). Results for $C_7H_8N_2O_2$ are in Appendix G. Although the difference in performance seems to be marginal, we stress that chemical validity is often determined by the last $10\%$ of the returns. Indeed, Table 1 and Fig. 9 in Appendix G highlight the higher quality of the structures generated by COVARIANT, indicating better generalization to unseen bags of larger size. OPT fails at this task.

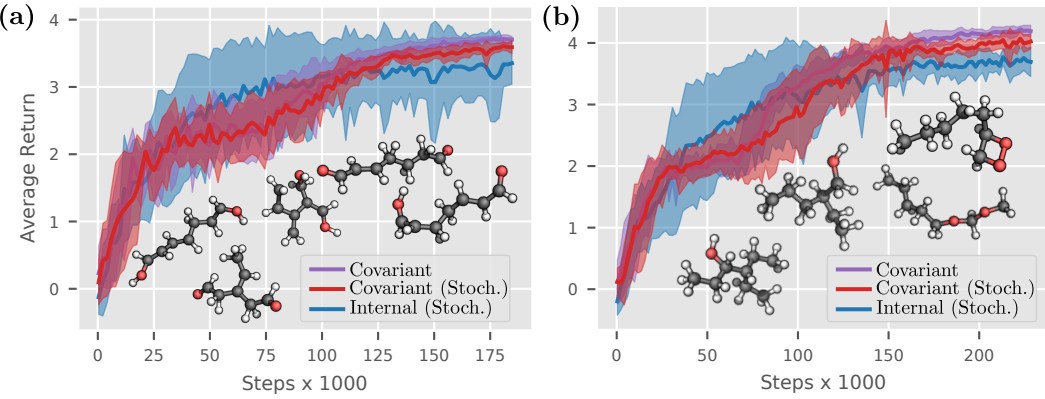

Figure 6: Average offline performance on the *stochastic-bag* task with $\mathcal{B}^* = C_7H_{10}O_2$ evaluated on (a) $C_7H_{10}O_2$ and (b) larger, unseen bags $\{C_6H_{14}O_3, C_7H_{16}O, C_7H_{16}O_2, C_8H_{18}O\}$ over 10 seeds. For comparison, we show an agent trained only on the test bags (purple). Error bars are two standard deviations. Molecular structures generated by COVARIANT (Stochastic) are shown.

## 6 CONCLUSION

We proposed a novel covariant actor-critic architecture based on spherical harmonics for designing highly symmetric molecules in 3D. We showed empirically that exploiting symmetries of the molecular design process improves the quality of the generated molecules and leads to better generalization. In future work, we aim to employ more accurate quantum-chemical methods (e.g., density functional theory) required for building transition metal complexes or structures in which weak intermolecular interactions are important. For that, however, the sample-efficiency of our agent needs to be improved. Finally, we aim to explore reward functions specifically tailored towards drug design.

## ACKNOWLEDGEMENTS

We thank A. J. Tripp and K. T. Jensen for useful discussions and feedback. RP receives funding from iCASE grant #1950384 with support from Nokia. JMHL acknowledges support from a Turing AI Fellowship under grant EP/V023756/1. This work has been performed using resources operated by the University of Cambridge Research Computing Service (funded by grant EP/P020259/1).

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

## A  REWARD CALCULATION

In the reward function, the energy $E$ has to be computed using quantum-chemical methods. For that, we use the fast semi-empirical Parametrized Method 6 (PM6) (Stewart, 2007). In particular, we use the implementation in the software package SPARROW (Husch et al., 2018; Bosia et al., 2020). For each calculation, a molecular charge of zero and the lowest possible spin multiplicity are chosen. All calculations are spin-unrestricted.

Limitations of semi-empirical methods are highlighted in, for example, recent work by Husch & Reiher (2018). More accurate methods such as approximate density functionals need to be employed especially for systems containing transition metals.

Further, we enforce that atoms are not placed too close ($< 0.6$ Å) nor too far away from each other ($> 2.0$ Å). If the agent places an atom outside these boundaries, the minimum reward of $-0.6$ is awarded and the episode terminates. Further, the environment encourages the agents to build single molecular structures by terminating the episode and return a reward of $-0.6$ if elements forming stable bimolecular compounds (e.g, $H_2$) are placed too far away from other atoms on the canvas.

## B  SPHERICAL HARMONICS

The spherical harmonics form an orthonormal basis of the Hilbert space of square-integrable functions $L_{\mathbb{C}}^2(\mathcal{S}^2)$. The first few spherical harmonics are given by:

$$Y_0^0(\vartheta, \varphi) = \frac{1}{2\sqrt{\pi}}, \tag{7}$$

$$Y_1^{-1}(\vartheta, \varphi) = \sqrt{\frac{3}{8\pi}} \sin\vartheta e^{-i\varphi}, \quad Y_1^0(\vartheta, \varphi) = \sqrt{\frac{3}{4\pi}} \cos\vartheta, \quad Y_1^1(\vartheta, \varphi) = -\sqrt{\frac{3}{8\pi}} \sin\vartheta e^{i\varphi}, \tag{8}$$

$$Y_2^{-2}(\vartheta, \varphi) = \sqrt{\frac{15}{32\pi}} \sin^2\vartheta e^{-i2\varphi}, \quad Y_2^{-1}(\vartheta, \varphi) = \sqrt{\frac{15}{8\pi}} \cos\vartheta \sin\vartheta e^{-i\varphi},$$

$$Y_2^0(\vartheta, \varphi) = \sqrt{\frac{5}{16\pi}} \left(3\cos^2\vartheta - 1\right), \tag{9}$$

$$Y_2^1(\vartheta, \varphi) = -\sqrt{\frac{15}{8\pi}} \cos\vartheta \sin\vartheta e^{i\varphi}, \quad Y_2^2(\vartheta, \varphi) = \sqrt{\frac{15}{32\pi}} \sin^2\vartheta e^{i2\varphi}.$$

The spherical harmonics are normalized such that:

$$\int_0^{2\pi} \int_0^{\pi} |Y_\ell^m(\vartheta, \varphi)|^2 \sin\vartheta d\vartheta d\varphi = 1 \qquad \forall \ell, m. \tag{10}$$

## C  CALCULATION OF INVARIANT FEATURES

One can obtain scalar invariats from the covariant features $\hat{f}$ (Anderson et al., 2019):

- Take the component $\ell = 0$: $\xi_1(\hat{f}) = \hat{f}_{\ell=0}$.

- Calculate the scalar product with itself: $\xi_2(\hat{f}_\ell) = \text{Re}[\tilde{\xi}_2(\hat{f})] + \text{Im}[\tilde{\xi}_2(\hat{f})]$, where $\tilde{\xi}_2(\hat{f}) = \sum_{m=-\ell}^{\ell} (-1)^m \hat{f}_\ell^m \hat{f}_\ell^{-m}$.

- Calculate the SO(3)-invariant norm: $\xi_3(\hat{f}_\ell) = \sum_{m=-\ell}^{\ell} \hat{f}_\ell^m \left(\hat{f}_\ell^m\right)^*$, where $*$ denotes the complex conjugate.

The invariant components are then concatenated $f^{\text{inv}} \leftarrow \mathcal{T}_{\text{inv}} = \xi_1(\hat{f}) \oplus \left(\bigoplus_{\ell=0}^{L} \xi_2(\hat{f}_\ell) \oplus \xi_3(\hat{f}_\ell)\right)$.

## D    PROBABILITY DISTRIBUTION FOR ORIENTATION

In the main paper, we propose the expression in Eq. (5) for the distribution $p(\tilde{x}|d, e, f, s)$. An alternative, equally valid expression is

$$p(\tilde{x}|d, e, f, s) = \left| \sum_{\ell=0}^{L_{\max}} \sum_{m=-\ell}^{\ell} \frac{1}{\sqrt{k}} \hat{r}_\ell^m Y_\ell^m(\tilde{x}) \right|^2, \qquad (11)$$

where the term $1/\sqrt{k}$ with $k = \sum_{\ell=0}^{L_{\max}} \sum_{m=-\ell}^{\ell} |\hat{r}_\ell^m|^2$ normalizes the distribution. We found experimentally that an agent using this expression performs worse when generating molecular structures featuring complex geometries. We hypothesize that this is because the distribution cannot get peaked enough for $L_{\max} \leq 5$. As larger $L_{\max}$ would result in a significant increase in computational complexity, we chose the expression in Eq. (5) over that in Eq. (11).

The normalization constant $Z$ in Eq. (5) is estimated via Lebedev quadrature with 1730 angular grid points (Lebedev, 1975; 1977). We sample from the distribution in Eq. (5) using rejection sampling (Bishop, 2009) with a uniform proposal distribution $q(\tilde{x}) = \frac{1}{4\pi}$. In rejection sampling, one first draws a sample from $\tilde{x}_0$ from $q(\tilde{x})$. Then, one generates a random number $u_0$ from the uniform distribution over $[0, Mq(\tilde{x}_0)]$, where $M$ is such that $Mq(\tilde{x}) \geq p(\tilde{x}|d, e, f, s)$. We determine $M$ by evaluating $p(\tilde{x}|d, e, f, s)$ on a uniform grid on $\mathcal{S}^2$ employing a Fibonacci 'sunflower' grid. Finally, if $u_0 > p(\tilde{x}|d, e, f, s)$ then the sample is accepted.

We ran an experiment to compare the COVARIANT agent as described in the main paper with an agent employing Eq. 11 for the distribution $p(\tilde{x}|d, e, f, s)$. In Fig. 7, the hypothesis that the distribution in Eq. 11 cannot get narrow enough is confirmed. After 40 000 steps, the online performance of the alternative agent COVARIANT (ALT.) converges to around 0.5, which is significantly lower compared to COVARIANT. Note that the difference is smaller when considering the offline return because the estimated global mode for each distribution could still be similar.

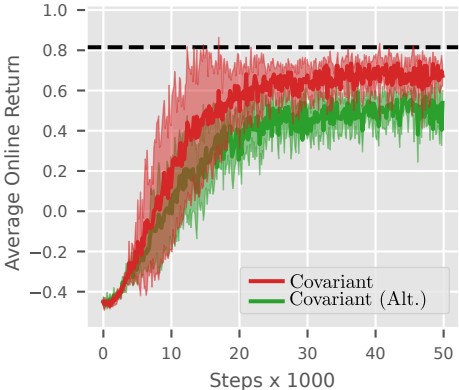

Figure 7: Comparison of COVARIANT agent (red) with an agent employing alternative distribution for the orientation $\tilde{x}$ (green). The average *online* performance on the *single-bag* task with $\mathcal{B} = \mathrm{SF}_6$ across 5 seeds is shown. Dashed lines denote the optimal return for the experiment. Error bars indicate two standard deviations.

## E    BASELINES

### E.1    OPT AGENT

Below, we detail the algorithm of the OPT agent. At the beginning of each experiment, the agent is given a canvas $\mathcal{C}_0$, a bag $\mathcal{B}_0$, and a black-box function that can compute the energy $E(\mathcal{C})$ and the atomic forces $F(\mathcal{C})$ for a given canvas. We assume a total charge of zero and a low-spin configuration. At the end of each experiment, we compute the total reward obtained for the final structure on canvas $\mathcal{C}_T$ and report the total number of energy and gradient computations.

1. If the canvas $\mathcal{C}_t$ is not empty, randomly choose a focal atom $f$ from the list of available atoms on the canvas. An atom is considered available if its number of neighbors is less than a predefined number that depends on its element (e.g., one for hydrogen and four for carbon). Two atoms on the canvas are neighbors if their Euclidean distance is below 1.5 Å. If there are no available atoms on the canvas, a focal atom is randomly chosen from the list of atoms on the canvas.

2. Randomly choose an element $e_t$ from the bag $\mathcal{B}_t$.

3. Randomly place the atom $a_t = (e_t, x_t)$ on a sphere with radial distance $d = 1.1$ Å around $x_f$ to obtain $\mathcal{C}_{t+1,\text{raw}}$. If the canvas is empty, place the atom at the origin.

4. Optimize only the position of $a_t$ using $F$ to obtain $\mathcal{C}_{t+1,\text{opt}}$.

5. Compute the energy difference $\Delta E(t) = E(\mathcal{C}_{t+1,\text{opt}}) - [E(\mathcal{C}_t) + E(\{e_t, \mathbf{0}\})]$.

6. If $\Delta E(t) > 0$, return $e_t$ to the bag and go back to step 1.

7. Optimize canvas $\mathcal{C}_{t+1,\text{opt}}$ using $F$ to obtain $\mathcal{C}_{t+1}$.

8. Increment $t$ by 1.

9. If the bag is not empty, go back to step 1.

In the experiments, the different agents need to be given a comparable computational budget to ensure a meaningful comparison of their performance. This is difficult as they use different computational resources: OPT runs on a CPU whereas INTERNAL and COVARIANT perform many of their computations on a GPU. However, we found experimentally that the quantum-chemical calculations are the most computationally expensive ones. These calculations are performed in the same way for all approaches. Therefore, we believe that by granting each approach the same number of PM6 calculations we achieve a fair comparison.

### E.2 OPTIMAL RETURN

The optimal return for the *single-bag* tasks was derived in the following way. First, we obtained molecular structures for the complexes SOF$_4$, IF$_5$, SF$_6$, and SOF$_6$. Subsequently, we performed a structure optimization using the PM6 method. Since the undiscounted return is path-independent, we determined the return $R(s)$ by computing the total interaction energy in the canvas $\mathcal{C}$, i.e.

$$R(s) = \left\{ \sum_{i=1}^{|\mathcal{C}|} E(\{e_i, \mathbf{0}\}) \right\} - E(\mathcal{C}). \tag{12}$$

## F EXPERIMENTAL DETAILS

### F.1 COMPUTING INFRASTRUCTURE

Experiments were run on an Intel Xeon E5-2650 v4 2.2GHz 12-core processor (96GiB RAM) and an Nvidia P100 GPU (16GiB). Our agent is implemented in the deep learning framework `PyTorch` (Paszke et al., 2019). Data analysis was performed with the Python libraries `matplotlib` (Hunter, 2007) and `pandas` (McKinney, 2010).

### F.2 IMPLEMENTATION DETAILS

The model architecture is summarized in Table 2, where the dimensions of $s^{\text{inv}}$ and $s^{\text{inv}}_{f,e}$ are $d^{\text{inv}} = (L_{\max} + 2) \cdot \tau \cdot 2$ and $d^{\text{inv}}_{f,e} = (L_{\max} + 2) \cdot \tau_e \cdot 2$, respectively. If possible, we made similar architectural choices as Simm et al. (2020), e.g. regarding the number of hidden units/layers, activation functions, and initialization schemes. We initialize the biases of each network with $0$ and each weight matrix as a (semi-)orthogonal matrix. After each hidden layer, a ReLU non-linearity is employed. As explained in the main text, both MLP$_f$ and MLP$_e$ use a masked softmax activation function to guarantee that only valid actions are chosen. To model the distance $d$, we employ a Gaussian mixture model consisting of $M = 3$ Gaussians. As we treat the standard deviations $\{\sigma_m\}_{m=1}^3$

Table 2: Model architecture for actor and critic networks.

| Network | Dimensions per layer | Output activation |
|---|---|---|
| $\text{MLP}_f$ | $d^{\text{inv}}, 128, 1$ | masked softmax |
| $\text{MLP}_e$ | $d^{\text{inv}}, 128, e_{\max}$ | masked softmax |
| MDN | $d^{\text{inv}}_{f,e}, 128, 6$ | linear $(\pi_m)$, tanh $(\mu_m)$ |
| $\text{MLP}_\phi$ | $d^{\text{inv}}, 128, 128$ | linear |
| $\text{MLP}_\rho$ | $128, 128, 1$ | linear |

as global parameters, the MDN has 6 outputs. Further, we rescale the means $\mu_m \in [-1, 1]$ to $\mu_m \in [d_{\min}, d_{\max}]$. If the sampled distance is negative, we clip the value at $0.001$.

Hyperparameters for CORMORANT are listed in Table 3. In our experiments, we found it important to use multiple filters $\tau_e$ per element (e.g. 4) and to set $L_{\max} = 4$. This gives the model enough flexibility to represent complex spherical distributions while remaining computationally tractable. For more details on CORMORANT, see the original work (Anderson et al., 2019). Further hyperparameters used in our experiments are in Table 4. PPO is known to be relatively robust with respect to the choice of hyperparameters, and we found the default values to be sufficient in most cases. Within the actor, the scaling parameter $\beta$ is important to avoid that the spherical distribution approaches a delta distribution. Note that values of $\beta$ can vary significantly across experiments and might require some tuning. Lastly, the right number of samples $S$ for the global mode estimation of the spherical distribution generally depends on the shape of the distribution. In particular, we would expect that more samples are required as the distribution becomes more peaked. Since we avoid pathological behaviors by scaling the distribution with $\beta$, we found $S = 1024$ to be sufficient for all our experiments.

Table 3: Hyperparameters for CORMORANT (Anderson et al., 2019) used in all experiments.

| Hyperparameter | Value |
|---|---|
| Number of Clebsch-Gordan layers | 3 |
| $L_{\max}$ in spherical harmonics series expansion | 4 |
| Number of filters $\tau_e$ per element | 4 |
| Number of filters $\tau$ | $\tau_e \cdot N_e$ |

Table 4: Hyperparameters for the *single-bag*, *multi-bag* and *stochastic-bag* tasks. Values in parentheses were only used for the *single-bag* task. For further details on how the PPO hyperparameters are defined, please refer to Schulman et al. (2017).

| Hyperparameter | Search Set | Value |
|---|---|---|
| Range $[d_{\min}, d_{\max}]$ (Å) | — | $[0.95, 1.80]$ |
| Number of workers | — | 10 |
| PPO clipping $\epsilon$ | — | 0.2 |
| PPO gradient clipping | — | 0.5 |
| PPO GAE parameter $\lambda$ | — | 0.95 |
| PPO value function coefficient $c_1$ | — | 1 |
| PPO entropy coefficient $c_2$ | $\{0.01, 0.05\}$ | 0.01 |
| Number of optimization epochs | — | 7 |
| Adam stepsize | — | $3 \cdot 10^{-4}$ |
| Discount factor $\gamma$ | — | 0.99 |
| Time horizon $T$ | — | $20 \cdot |\mathcal{B}|$ |
| Distance clipping | — | 0.001 |
| Scaling factor $\beta$ | $\{-100, -10, -1, 1, 10, 100\}$ | 100 $(-10)$ |
| Number of samples $S$ for mode estimation | — | 1024 |

## G   ADDITIONAL RESULTS

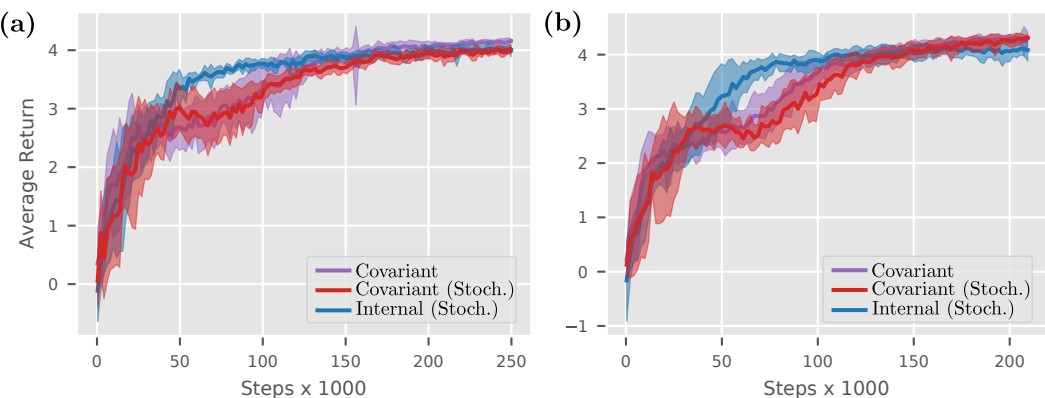

Figure 8: Average offline performance on the *stochastic-bag* task with $\mathcal{B}^* = C_7H_8N_2O_2$ evaluated on (a) $C_7H_8N_2O_2$ and (b) larger, unseen bags $\{C_6H_{14}O_3, C_7H_{16}O, C_7H_{16}O_2, C_8H_{18}O\}$ across 10 seeds. For comparison, we show an agent that is trained only on the test bags (purple). Error bars indicate two standard deviations.

In Fig. 9, a selection of molecular structures generated by the three agents during offline evaluation is shown. The agents are trained on a distribution over bags with $\mathcal{B}^* = C_7H_{10}O_2$ and tested on sets of larger, out-of-distribution bags $\{C_6H_{14}O_3, C_7H_{16}O, C_7H_{16}O_2, C_8H_{18}O\}$ (cf. Fig. 6 in the main text). From visual inspection of the structures, it can be seen why OPT fails at this task: it tends to generate molecular clusters, often containing $H_2$. Similarly, INTERNAL commonly builds $H_2O$ molecules instead of constructing a single organic molecule out of the atoms in the bag. By contrast, COVARIANT often builds valid molecules. Further, its generated structures are more often "branched" than those of INTERNAL, indicating a higher degree of complexity.

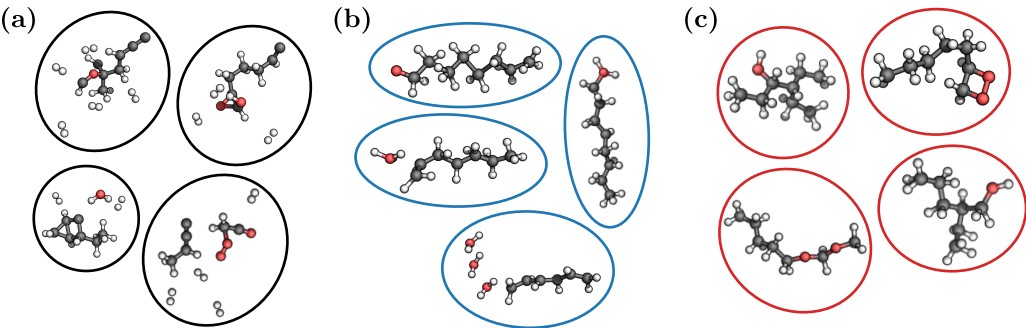

Figure 9: Selection of molecular structures generated by (a) OPT, (b) INTERNAL, and (c) COVARIANT during the last offline evaluation. The agents are trained on a distribution over bags with $\mathcal{B}^* = C_7H_{10}O_2$ and tested on the out-of-distribution bags $\{C_6H_{14}O_3, C_7H_{16}O, C_7H_{16}O_2, C_8H_{18}O\}$.

Next, we assess the ability of COVARIANT to generate solvation clusters—a type of molecular structure that cannot be built with graph-based approaches. Following Simm et al. (2020), we task the agent to place 5 water molecules around a formaldehyde molecule that is already on the canvas at the beginning of each episode. In addition, the reward function is augmented with a penalty term for placing atoms far away from the center, i.e. $r(s_t, a_t) = -\Delta_E - \rho\|x\|_2$, where $\rho$ is a hyper-parameter that is set to $0.01$ (see Simm et al. (2020) for details). Therefore, the agent needs to place the water molecules such that hydrogen bonds can be formed between water molecules and between water molecules and the solute.

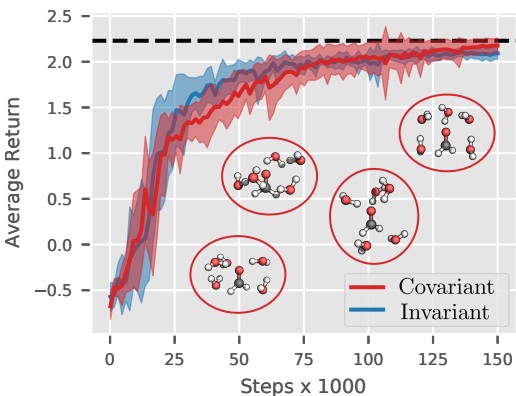

Figure 10: Average offline performance on the *solvation* task with 5 $H_2O$ molecules and formaldehyde as the solute across 10 seeds. Error bars show two standard errors. The dashed line denotes the optimal return. A selection of molecular clusters generated by the COVARIANT agent is shown.

From Fig. 10, it can be seen that COVARIANT can solve this task by constructing stable $H_2O$ molecules and placing them in the vicinity of the solute. From visual inspection of the generated structures, it can be observed that in many cases COVARIANT arranges the molecules such that intermolecular bonds can be formed. However, it should be noted that the quantum-chemical method used in the reward function is not very well suited for modeling these interactions. Finally, Fig. 10 shows that while INTERNAL learns faster at the beginning of training, COVARIANT is slightly outperforming INTERNAL towards the end.

## H   RUNTIME EVALUATION

We compared the runtimes between OPT, INTERNAL, and COVARIANT. For instance, for the *single-bag* task with the bag $C_3H_5NO_3$, $T = 240$ steps of the last rollout took COVARIANT and INTERNAL on average 12 and 11 seconds (s), respectively. The final offline evaluation took the agents on average 4 and 1s, respectively. This speed difference is mainly due to the relatively slow rejection sampling procedure in COVARIANT. Each iteration, policy optimization took on average 2 and 6s for the agents COVARIANT and INTERNAL, respectively. In this case, INTERNAL is slower than COVARIANT as it performed around twice as many epochs during optimization due to early stopping. Since there is no training for OPT, this agent was overall faster than the others. Further, we note that the largest fraction of time was spent on the quantum-chemical calculations which are the same for all agents. The time the quantum-chemical calculation takes to converge depends not only on the size but also on the geometry of the input structure. The entire experiment took COVARIANT approximately 4 hours, INTERNAL 5 hours, and OPT 3 hours.

