# OpenReview forum: "Symmetry-Aware Actor-Critic for 3D Molecular Design"
_ICLR.cc/2021/Conference — ICLR 2021 Poster_

### Official Review · AnonReviewer2 · 2020-10-25
**Elegant theoretical justification, but experimental performance is lacking**

**Rating:** 6
**Confidence:** 4

**Review:**

This work presents an approach for 3D molecular design using reinforcement learning that exploits rotational symmetries in molecular conformations. The formulation involves an MDP that selects atoms from a “bag” and positions them in 3D space. The reward function is based on PM6 energies to encourage the generation of low-energy structures. The manuscript is well-written and provides ample context and appropriate references.

The generative formulation directly builds on that of Simm et al. 2020 to choose a focal atom, choose an element to add, choose a distance, then choose an orientation. The consideration of rotational covariance during new atom placement and the use of Cormorant (Anderson et al. 2019) for embedding seem to be the primary novelties of this work. Thus, Simm et al.’s approach relying on an internal coordinate representation serves as the primary baseline in the empirical evaluations here; “Opt” is meant to serve as an essentially-random baseline and performs predictably poorly. One could even envision using a 2D generative model with a heuristic conformer generation scheme followed by geometry optimization using PM6 forces as a stronger baseline approach.

The benefit of Covariant over Internal in terms of actual optimization performance (Figure 5, Figure 6) appears marginal. Did the authors perform any experiments to optimize more than just PM6 energy (e.g., a computed property) to compare the two methods? It’s not clear how significant the observed differences are and to what extent achieving rotational covariance helps in this task. An ablation study that decouples the effects of the rotationally-invariant critic and the rotationally-covariant actor could be informative. How sensitive are either of these models to hyperparameter selection?

Covariant does appear to generate a greater number of valid structures (Table 1), although this evaluation maps the 3D coordinates back to a SMILES string to check for validity and uniqueness, which seems not to be the intended application of this work. For that evaluation, one could consider using purely graph- or SMILES-based approaches and more traditional generative methods.

A primary advantage of this formulation seems to be the ability to generate molecular clusters (in contrast to 2D approaches) and periodic materials (in contrast to Simm et al.). The current set of experiments do not demonstrate this ability in my opinion. There also is not evidence that “building in this inductive bias enables [one] to generate geometrically complex molecular structures that were unattainable with previous approaches.”

The integration of rotational covariance with 3D molecular generation is elegant and, to my knowledge, novel. I don’t disagree with the premise that one *should* strive to have these invariances, but I don’t believe that the current set of experiments actually demonstrate the superiority.

---

> ### Author Response · Authors · 2020-11-19
> **Response to Reviewer 2**
>
> Thank you very much for reviewing our paper. We are glad you found the manuscript to be well-written and our theoretical justification to be elegant. Based on your feedback, we have significantly expanded the experimental section. Below we respond to your comments and suggestions in more detail.
>
> **Baselines**
> We chose Opt as a baseline because it is, in principle, able to solve all tasks. This is in contrast to any 2D generative model, which would fail to solve the tasks in Fig. 5 (previously Fig. 4). Indeed, we find that Opt does perform well on some tasks: it found the optimal structures for all four bags in Fig. 5, and achieved the highest diversity for C4H7N in Table 1. We have updated the main text to better motivate the choice of baselines.
>
> **Advantage over Simm et al.**
> For the systems studied here, chemical validity is often determined on the last 10% of the optimal return. Therefore, small differences in the return can have a significant effect on the validity of the generated molecules. While Internal and Covariant have comparable learning curves, Table 1 shows that Covariant generates significantly more valid molecules. Moreover, the now reported RMSD values in Table 1 suggest that Covariant also generally builds slightly more stable molecules. Lastly, we have added a qualitative comparison of the molecular structures generated by all agents in Appendix G, highlighting the novel agent's superiority.
>
> **Going Beyond Energy Optimization**
> It is our goal to go beyond building stable molecules and to amend the reward function to guide the agent towards building molecules featuring some desirable property. With our newly added results on the solvation-task (see Appendix G) from Simm et al., we show in a proof-of-principle study that this is indeed possible. It can be seen that the agent successfully learns to place water molecules closely around a given solute through a slight modification of the reward function. Optimizing for molecular properties, such as solvability, is left for future work.
>
> **Ablation Study**
> Could you please further clarify your suggestion? It is not quite clear to us how an informative ablation study would look like in this case. That is because decoupling the actor from the critic is difficult as they share the same latent state representation. Moreover, it is unclear what would be sensible replacements for the ablated networks.
>
> **Sensitivity to Hyperparameters**
> We have expanded the discussion on hyperparameter choices in Appendix F.2 with a focus on hyperparameter sensitivity. In summary, we found it important to use multiple filters $\tau_e$ per element (e.g. $4$) and to set $L_{\text{max}} = 4$. This gives the model enough flexibility to represent complex spherical distributions while still remaining computationally tractable. Within the actor, the scaling parameter $\beta$ is important to avoid that the spherical distribution approaches a delta distribution. Lastly, we made similar architectural choices as Simm et al. if possible, e.g. regarding the number of hidden units/layers, activation functions and initialization schemes of the actor and critic networks.
>
> **Evaluation of Validity and Diversity**
> We are mainly interested in solving tasks that go beyond what purely graph-based methods can solve by design. This includes generating molecules with high coordination numbers such as SF6 or SOF4 (see Fig. 5 in the updated manuscript), solvation clusters (newly added, see Fig. 10), and transition metal complexes and (amorphous) solids (left for future work). However, since string and graph representations are not well-suited for designing such molecules, it is difficult to directly compare to most prior work. To still enable comparisons, in Table 1 we follow the Guacamol benchmark and report the validity, diversity, and stability of molecules that would be feasible. In the updated manuscript, we have clarified the purpose of this evaluation.
>
> **Comparison with Graph-Based Approaches**
> The main advantage of this work compared to both 2D approaches and the approach by Simm et al. is that our agent is able to build molecules featuring high symmetry and coordination numbers beyond what is typically found in organic chemistry, e.g. trigonal bipyramidal, square pyramidal, and octahedral symmetries. We believe the experiments in Fig. 5 convincingly demonstrate the benefits of our approach compared to Simm et al. Further, note that no string- or graph-based method would be able to build such molecules. We experimentally verified this using RDKit, which cannot handle these structures. In the updated manuscript, we have clarified this by stating our claims more precisely, and by expanding the discussion of the experimental results. Finally, we have added a solvation experiment that further highlights the benefits compared to graph-based methods.

---

### Official Review · AnonReviewer4 · 2020-10-27

**Rating:** 6
**Confidence:** 4

**Review:**

### Summary of the paper
The paper utilizes a actor-critic framework for 3D molecular design. The central part of the approach is a rotation equivariant network (Comorant). For each atom, Comorant learns a state representation so that it is equivariant under rotation. The method is evaluated on 9 different molecules and it outperforms Simm et al.'s method in terms of validity and diversity.

### Strength and weakness
1. The method adopts a better model architecture (Comorant) to learn the state representation. However, Comorant is developed by Anderson et al. Therefore it cannot be counted as the contribution of this paper. The actor-critic formulation is also standard in RL. The sequential decision process is also similar to Simm et al. The technical innovation is not original enough.

2. The evaluation protocol is problematic. In table 1, authors only report validity and diversity of generated compounds. The validity is defined as "successfully parsed by RDKit". To my knowledge, RDKit validity checking is based on 2D constraints such as valence, aromaticity and kekulization. It does not capture 3D information at all. Are the generated compound stable? What is the RMSD of generated compounds? Simm et al. 2020 reported RMSD to measure the structural stability. Why RMSD metric is missing? This is important because 2D graph generation models already satisfy validity quite well (100% validity for even large molecules).

3. Motivation is not clear. In the paper, authors state that the choice of focal atom, element and distance have to be *invariant* to rotation. It seems like invariant representation is sufficient. Moreover, the advantage over Simm et al. is not clear in section 4. Why highly symmetric states are problematic for prior work? How does this method solves this issue? I am afraid that the experiment section does not address this, since most test cases are not "highly symmetric" to my knowledge.

### Overall evaluation
I vote for rejection. My major reason is the problematic evaluation protocol. RMSD must be added to evaluate the stability of compounds. Moreover, I would like to see evaluation on "highly symmetric" cases. I think it's important since authors state that this is the major limitation of prior work.

### Question
1. Is the method scalable to large molecules? How is the runtime of your model compared to Simm et al.?

### Post rebuttal feedback
It's good to see that RMSD experiments are added and the results are better than Simm et al. Therefore, I am raising my score to 6. I also realized that the validity calculation is different from standard graph generation methods. The validity results now look reasonable to me.

---

> ### Author Response · Authors · 2020-11-19
> **Response to Reviewer 4**
>
> Thank you very much for reviewing our paper. Based on your feedback, we have made significant improvements to the manuscript. Below we respond to your comments and concerns in more detail.
>
> **Contributions**
> We have expanded the introduction in the main text to further highlight our contributions. Our main technical contribution is to integrate a covariant state representation into a novel actor-critic network architecture for building molecular structures directly in Cartesian coordinates. As pointed out by Reviewer 1, the proposed architecture is the first model to generate 3D molecules that uses covariant features to leverage the rotational symmetries of the design process. In contrast to Cormorant, we do not predict scalars such as the electronic energy, but instead output spherical harmonics coefficients, which are then used to model a generally multi-modal distribution on the sphere. We stress that integrating such a distribution into a stochastic policy is challenging. For example, one has to be careful about how to condition this distribution on previous sub-actions (e.g. through a Clebsch-Gordan product) without breaking rotational covariance. Further, it is not trivial to sample from this distribution (which is required to perform exploration) or evaluate the global mode (which is required to perform offline evaluation).
>
> **Evaluation Protocol**
> To expand our evaluation metrics, we have added the RMSD to Table 1 as a measure of the stability of the generated structures. As reported in the updated manuscript, we find that for most experiments the generated molecules by our agent are more stable compared to Simm et al. Further, we would like to clarify that to compute the validity and diversity of the generated molecules, one must generate a 2D graph from a 3D structure through a process that does take atomic distances (and therefore 3D information) into account.
>
> **Importance of Covariance**
> While the choice of the focal atom, element, and distance are invariant to rotation, the orientation needs to be covariant to rotation. Therefore, a purely invariant representation is not sufficient. We guarantee rotational covariance by modeling the orientation through a spherical distribution. We define this distribution through a spherical harmonics series expansion, where the coefficients are predicted by Cormorant. In contrast, Simm et. al achieved rotational covariance by modeling the position of the next atom using already placed atoms as reference points. When the structure is highly symmetric, this representation fails as one can no longer uniquely assign these reference points. This problem causes the approach by Simm et al. to fail in Fig. 5 (previously Fig. 4). We have updated Section 4 (Related Work) to better motivate our paper and clarify differences to prior work.
>
> **Experiments with Highly Symmetric Structures**
> The molecules built in Fig. 5 feature high symmetry and coordination numbers beyond what is typically found in organic chemistry, e.g. trigonal bipyramidal, square pyramidal, and octahedral symmetries. We believe these experiments convincingly demonstrate the benefits of our approach compared to Simm et al. Further, no string- or graph-based method would be able to build such molecules. We have updated the experimental section to clarify this.
>
> **Scalability and Runtime**
> As shown in Table 1 and Fig. 6, our agent is able to build a diverse range of stable molecules consisting of up to 28 atoms. In contrast, the agent from Simm et al. struggles to design valid molecules of such size. Further, we have added a runtime comparison for the single-bag task with the bag C3H5NO3. The entire experiment took Covariant approximately 4 hours, Internal 5 hours, and Opt 3 hours. We have summarized our findings in Appendix H.

---

> > ### Comment · AnonReviewer4 · 2020-11-21
> > **Response**
> >
> > Thank you for the response. Most concerns has been addressed and I have raised my score to 6. I think it will be great if you can clarify how you convert generated 3D structure into a 2D graph (and SMILES string). Based on author's definition, I believe the notion of "validity" here is slightly different from standard molecular graph generation. Please add more description in the paper to make it crystal clear (e.g., show the procedure of 3D -> 2D conversion).

---

> > > ### Author Response · Authors · 2020-11-23
> > > **Graph Generation**
> > >
> > > Thank you for your response.
> > > We now clarify in the manuscript that a generated structure is considered valid if it can be successfully converted to a molecular graph with the tool XYZ2Mol [1, 2].
> > > Then, two molecules are considered identical if their molecular graphs yield the same SMILES strings under RDKit.
> > > Thus, once we have obtained a graph representation of the generated structure, the notion of validity is the same as in graph-based approaches.
> > >
> > > [1] Kim, Y.; Kim, W. Y. Universal Structure Conversion Method for Organic Molecules: From Atomic Connectivity to Three-Dimensional Geometry. Bull. Korean Chem. Soc. 2015, 36 (7), 1769–1777. https://doi.org/10.1002/bkcs.10334.
> > > [2] Jensen, J. XYZ2Mol https://github.com/jensengroup/xyz2mol (accessed Sep 12, 2019).

---

### Official Review · AnonReviewer1 · 2020-10-28
**Well-motivated approach to 3d molecular structure generation**

**Rating:** 8
**Confidence:** 4

**Review:**

The paper proposes an actor-critic neural network architecture for autoregressive generation of 3D molecular structures with reinforcement learning (RL). It builds upon the RL approach by Simm et al. (2020) which makes use of internal coordinates in order to deal with the symmetries that occur when placing atoms in the molecular design process.
The main contribution is the introduction of a covariant state-action representation extracted with Cormorant (Anderson et al., 2019) and a covariant, spherical distribution for the orientation of newly placed atoms (w.r.t. a focal atom) replacing the internal coordinates of the former approach. It is shown that these architectural and conceptual changes allow to build highly symmetric structures (which were problematic with the former approach) and improve the generalization as well as the quality of generated molecules in several tasks from the RL environment MolGym.

The paper has a clear structure, is well-written and includes several experiments to verify the main claims. To the best of my knowledge, the proposed architecture is the first model for generation of 3d molecules that uses covariant features to leverage the rotational symmetries of molecules. This seems to improve the model by Simm et al. (2020), especially when it comes to symmetric structures such as $SF_6$. The proposed covariant formulation could be beneficial for other existing or future generative models for 3d molecules, as well.

The experimental section could be improved by additional experiments to allow for comparison between the proposed architecture and other approaches for generation of molecular graphs and 3d molecules (Liu et al , Gebauer et al. 2019).  In many real-world molecular design tasks, a variety of (possibly diverse) molecules that exhibit certain target properties are desirable. It would be interesting to see whether the proposed model can also be applied for these purposes instead of the rather limited task to generate stable molecule from a predefined bag.

Pros
----
- well motivated neural network design incorporating the necessary symmetries
- extended experiments show generalization performance

Cons
----
- It does not become quite clear, but as far as I understand only the most probable action is chosen during offline evaluation (Sec 3.3). An additional experiment on how the validity and diversity depend on the degree of exploration would be useful.
- The ms could be improved by additional experiments that allow for comparison with previous molecule generation models on different tasks (see Liu et al 2018, Mansimov et al 2019, Gebauer et al 2019).

Update:
I raised my score to reflect the added metric and experiments that improved the paper.

---

> ### Author Response · Authors · 2020-11-19
> **Response to Reviewer 1**
>
> Thank you very much for reviewing our paper. We appreciate that you find our work to be well-motivated and our extended experiments showing generalization performance to be convincing. Below we respond to your suggestions to further improve the paper.
>
> **Performance of Exploration Policy**
> You are right that only the most probable action is chosen during offline evaluation. This is standard practice in RL. In a new experiment, we study the effect of exploration on the validity and diversity of generated structures. For that, we compared the validity and diversity scores obtained from a final online and offline evaluation of an agent trained on the bag C5H3NO3. Averaged over 10 seeds, we found that exploration causes the validity to drop from 90% to 50% and the diversity to increase from 5 to 21.
>
> **Comparison with Prior Work**
> To further facilitate the comparison with prior work, we have added the RMSD metric assessing the stability of generated structures. Further, we now critically compare our agent with supervised graph-based approaches such as Liu et al. (2018) in terms of validity and diversity and highlight the fundamental differences between the approaches. Mansimov et al. (2019) present a probabilistic model for structure prediction given a molecule's graph representation. Since this a significantly different task, we forego a direct comparison.
>
> **Building Molecules with Target Properties**
> It is in the spirit of this work to go beyond building stable molecules by modifying the reward function to guide the agent towards building molecules featuring desirable properties (e.g., drug-likeliness). To some extent, we do this in our newly added results on the solvation-task (see Appendix G). In this task, the agent successfully learns to place water molecules closely around a given solute through a slight modification of the reward function.

---

### Author Response · Authors · 2020-11-19
**General Response**

We thank all reviewers for their time and effort in reviewing our paper. We are glad you found that our paper "has a clear structure, is well-written and includes several experiments to verify the main claims" (R1), our work to be "elegant and [...] novel" (R2), and that our method "outperform[s] Simm et al.'s method in terms of validity and diversity" (R4).

In summary, we have updated our manuscript with the following changes:
- we added the RMSD evaluation metric to measure the stability of generated molecules and expanded comparison with prior work
- we added qualitative results on generated molecules and ran additional experiments on building molecular clusters to highlight the benefits of our approach
- we clarified our technical contributions and stressed the importance of a covariant state-action representation
- we added further details on hyperparameter sensitivity and runtimes

Below we respond to the reviewers' comments individually.

---

### Decision · Program_Chairs · 2021-01-07
**Final Decision**

**Decision:**

Accept (Poster)

**Comment:**

The paper provides a new covariant approach to 3D molecular generation motivated by the desire handle compounds with symmetries. To this end, the method uses equivariant state representations for autoregressive generation, built largely from recently proposed covariant molecular networks (comorant), and integrating such representations within an existing actor-critic RL generation framework (Simm et al). The selection of focal atom, element to add, and the distance are realized in an equivariant manner while the compound valuation remains invariant to rotation. The approach is clean and well-executed. The authors added additional experiments (e.g., RMSD demonstrating stability of generated compounds) to further reinforce the case for the method.